# Mechanisms Underlying the C₃–CAM Photosynthetic Shift in Facultative CAM Plants

**Shuo Qiu, Ke Xia, Yanni Yang, Qiaofen Wu and Zhiguo Zhao \***

Guangxi Key Laboratory of Plant Functional Phytochemicals and Sustainable Utilization, Guangxi Institute of Botany, Guangxi Zhuang Autonomous Region and Chinese Academy of Sciences, Guilin 541006, China
\* Correspondence: zwskfc@139.com; Tel.: +86-773-3550103; Fax: +86-773-3550067

**Abstract:** Crassulacean acid metabolism (CAM), one of three kinds of photosynthesis, is a water-use efficient adaptation to an arid environment. CAM is characterized by $CO_2$ uptake via open stomata during the nighttime and refixation $CO_2$ via the Calvin cycle during the daytime. Facultative CAM plants can shift the photosynthesis from $C_3$ to CAM and exhibit greater plasticity in CAM expression under different environments. Though leaf thickness is an important anatomical feature of CAM plants, there may be no anatomical feature changes during the $C_3$–CAM transition for all facultative CAM plants. The shift from $C_3$ photosynthesis to CAM in facultative CAM plants is accompanied by significant changes in physiology including stomata opening, $CO_2$ gas exchange and organic acid fluxes; the activities of many decarboxylating enzymes increase during the shift from $C_3$ to CAM; the molecular changes occur during the photosynthesis $C_3$–CAM shift involved DNA hypermethylation, transcriptional regulation, post-transcriptional regulation and protein level regulation. Recently, omics approaches were used to discover more proceedings underling the $C_3$–CAM transition. However, there are few reviews on the mechanisms involved in this photosynthetic shift in facultative CAM plants. In this paper, we summarize the progress in the comparative analysis of anatomical, physiological, metabolic and molecular properties of facultative CAM plants between $C_3$ and CAM photosynthesis. Facultative CAM plants also show the potential for sustainable food crop and biomass production. We also discuss the implications of the photosynthesis transition from $C_3$ to CAM on horticultural crops and address future directions for research.

**Keywords:** $C_3$ photosynthesis; crassulacean acid metabolism (CAM); shift mechanisms; facultative crassulacean acid metabolism (CAM) plants; environments





## 1. Introduction

In the plant kingdom, there are three kinds of photosynthetic pathways: $C_3$, $C_4$ and crassulacean acid metabolism (CAM). CAM is characterized by $CO_2$ uptake during the nighttime via open stomata, when $CO_2$ is combined with phosphoenolpyruvate (PEP) and stored as organic acids (mainly malic acid). Then, organic acids are decarboxylated in the vacuoles during daytime and $CO_2$ is refixed via the Calvin cycle [1,2]. Some plants can switch their photosynthesis between $C_3$ and CAM, which are referred to as facultative (inducible or $C_3$/CAM intermediate) CAM plants. The first discovered facultative CAM plant was *Mesembryanthemum crystallinum*, the common iceplant [3]. Other well-known facultative CAM plants include *Sedum album* [4], *Clusia minor* [5], *Talinum triangulare* [6], etc. These species are sampled as models to study other facultative CAM plants, as they can shift their photosynthetic mode in response to water deficit and other environmental stressors [7,8]. Furthermore, recent evidence suggests that facultative CAM plants may be more widespread among vascular plants than previously thought [9].

As we all know, photosynthesis is one of the most important chemical reactions on the earth [10]. Crassulacean acid metabolism (CAM), with higher photosynthetic, water-use and possibly nutrient-use efficiency, represents higher carbon-concentrating

mechanisms (CCMs) than $C_3$ photosynthesis in response to a warmer and drier world [11]. The phenomena of CAM is a typical ecophysiological adaptation to arid conditions [12–14]. Expression of CAM modes includes obligate CAM, facultative CAM, CAM-idling and CAM-cycling. Facultative CAM mode is one of the four plastic expressions of CAM modes. Facultative CAM plants can struggle with variable environments through the facultative CAM mode [14]. Recently, engineering CAM-related genes to $C_3$ crops to improve water-use efficiency (WUE) has caused extensive attention [15–17]. In order to engineer CAM into $C_3$ crops, a deep understanding of CAM-related genes and metabolic pathways is urgently needed [18].

However, the shift mechanism underlying $C_3$ photosynthesis to CAM in facultative CAM plants is complicated, concerning the genetic changes required for the progression and reversion of this shift [19]. In order to engineer CAM into $C_3$ crops to increase the water-use efficiency (WUE), a few facultative CAM species (such as *M. crystallinum*) were regarded as key tools to identify the genes involved in the CAM pathway and their respective regulation mechanisms [11,15,20,21]. In this paper, we review the signaling stress factors inducing $C_3$-photosynthesis to CAM in facultative CAM plants and assess progress in the analysis of anatomical, physiological, metabolic and molecular differences for facultative CAM plants between the $C_3$ and CAM mode. We also review their implications on horticultural crops and address directions for future research.

## 2. Signaling Stress Factors

CAM is a plastic photosynthetic adaptation found in plants in abiotic stress environments (such as drought, salinity, extreme temperature, etc.) [13]. Environmental, hormonal and circadian changes can regulate the CAM expression in facultative CAM species [22]. In facultative CAM species, photosynthesis can switch from $C_3$ to CAM modes after induction by abiotic stress, such as atmospheric $CO_2$ concentration, drought, salinity, photoperiod and light [12,23–25]. For instance, *M. crystallinum* switches its photosynthetic mode from $C_3$ to CAM under water or salinity stress [7]. Light intensity and quality also play a crucial role for the $C_3$–CAM transition [12,26]. All these signaling factors connect via a closed network and directly or indirectly affect each other [2].

The exogenous application of ethylene or abscisic acid (ABA) could induce the $C_3$–CAM transition in a few facultative CAM species [27]. The degree of CAM expression was positively correlated with ABA and trans-zeatin, but negatively correlated with cytokinins and jasmonic acid (JA) [28,29]. Exogenous hydrogen peroxide ($H_2O_2$) and root signaling also could induce the $C_3$–CAM transition in *M. crystallinum*, respectively [30–32].

## 3. Anatomical Variations during the $C_3$–CAM Shift in Facultative CAM Plants

Leaf thickness is an important anatomical feature for CAM plants. CAM is often associated with succulent leaves; indeed, the tissue succulence of CAM species has been observed in many plant families, such as Crassulaceae, Orchidaceae and Clusiaceae [33–36]. Many arid CAM plants with succulent nature are beneficial as they store more water than $C_3$ and $C_4$ species [37]. A study reported, by analyzing the leaf thickness and leaf $\delta^{13}C$ values in 173 tropical orchids, that the leaf was the thickest in the strong CAM species [36]. However, some plants with thinner leaves can also fix $CO_2$ through the CAM pathway; for example, *Dendrobium bigibbum* (a CAM orchid) can yield $\delta^{13}C$ with $-11.9‰$, despite the leaf thickness being only 0.79 mm [34], which proved that the assumption about obligate CAM species possessing more succulent leaves than facultative CAM species was not accurate [38]. In *M. crystallinum*, leaf succulence increased during the $C_3$ to CAM transition after 5 days of the salt treatment [39], but some CAM species (e.g., bromeliads) do not have succulent photosynthetic organs [40,41]. Recently, it was also hypothesized that the evolution of facultative CAM plants did not require major changes in anatomy [42,43]. Winter thought that strong CAM plants needed significant anatomical modifications, whereas facultative or weak CAM plants may not require them, suggesting there may be no anatomical feature changes during the $C_3$–CAM transition in facultative CAM plants [9]. Investigations showed that

leaf anatomy was not correlated to CAM function in *Yucca gloriosa* (facultative CAM species, a $C_3$+CAM hybrid species) [44], and the relationships between leaf anatomy and degree of CAM expression were not very close [45]. Herrera [39] reported that less succulence is not a typical feature for facultative CAM plants. Thus, the leaf thickness, as an indicator, cannot completely distinguish between plant species, suggesting that the relationship between anatomical leaf features and CAM expression requires further investigation.

That being said, there may be some changes in vacuole and chloroplast anatomy during the photosynthesis shift from $C_3$ to CAM. Malic acid accumulates and releases in the vacuole. More particularly, the fluidity of the tonoplast will reduce after the photosynthesis shift from $C_3$ to CAM, in turn decreasing the vacuolar mobilization of malic acid [46], indicating that vacuole size may increase before the shift to CAM in the leaves [47]. Chloroplasts, the main sites of photosynthesis in plants, can regulate the facultative CAM plants to acclimate to high salinity environments [48], and show a severe thylakoid swelling at midday in CAM plants [49]. However, whether such changes in the ultrastructure level also occur in other facultative CAM species remains unclear. Young leaves in obligate CAM plants take up $CO_2$ by $C_3$ photosynthesis, while mature leaves take up $CO_2$ by CAM. It is worth exploring the changes in ultrastructure level (i.e., vacuole and chloroplast) that occur throughout the development process of obligate CAM plants.

## 4. Physiological Mechanisms during the $C_3$–CAM Shift in Facultative CAM Plants

Compared to $C_3$ and $C_4$ metabolism, CAM is characterized by $CO_2$ uptake at night and stomatal closure during the day. For facultative CAM species, the $C_3$ photosynthesis pathway is used to fix $CO_2$ when the plants are under well-conditions. However, the photosynthesis can shift to the CAM mode once induced by abiotic stress; stomata open at night and atmospheric $CO_2$ is absorbed by phosphoenolpyruvate carboxylase (EC 4.1.1.31; PEPC) via oxaloacetate into malic acid. On the following day, stomata close and malic acid is released from the vacuoles and decarboxylated, while $CO_2$ is refixed by the Calvin cycle [1]. Therefore, in facultative CAM plants, the $CO_2$ gas exchange exhibits a difference between daytime and nighttime, as well as the changes in the activities of PEPC and Rubisco, which are important features to distinguish the $C_3$ from the CAM mode. Figure 1 shows the physiological and metabolic shift of the facultative CAM plants from $C_3$ to CAM photosynthesis.

Since this $C_3$ to CAM transition was first recognized, *M. crystallinum* was used as a model plant to examine the associated changes in enzyme activities [50]. PEPC, one of the key enzymes, is involved in primary carboxylation during both CAM and $C_4$ photosynthesis. PEPC is widely distributed across plants, algae and bacterial species; it catalyzes the irreversible β-carboxylation of phosphoenolpyruvate (PEP) in the presence of $HCO_3^-$ to yield oxaloacetate (OOA) and Pi. In CAM and $C_4$ photosynthesis, this enzyme is responsible for the primary fixation of inorganic carbon. PEPC also is a major anaplerotic enzyme in most non-photosynthetic organs and the leaves of $C_3$ plants [51,52]. Furthermore, the activities of many decarboxylating enzymes increase during the shift from $C_3$ to CAM, such as the cytosolic NADP-malic enzyme (NADPME, EC 1.1.1.40), the mitochondrial NAD-malic enzyme (NADME, EC 1.1.1.38) and PEP carboxykinase (PEPCK). This increased enzymatic activity is thought to be an indicator of the start of CAM photosynthesis [22,50].

In the plant, reactive oxygen species (ROS), including the superoxide radicals ($O_2^-$), the hydroxyl radical (OH) and hydrogen peroxide ($H_2O_2$), are always formed in response to environmental stress [53,54]. Oxidative stress could lead to the photosynthesis switch from $C_3$ to CAM in facultative CAM plants [55,56]. Enzymatic antioxidants and non-enzymatic antioxidants are involved to protect the plants from oxidative damage by the scavenging of ROS [57]. During the $C_3$–CAM shift in facultative CAM plants (e.g., *Sedum album*) induced by water stress, antioxidative enzymes, such as superoxide dismutase (SOD), peroxidase (POD), ascorbate peroxidase (APX), catalase (CAT), etc. [4,58,59]. In *M. crystallinum*, FeSOD activity increases more rapidly during the first few days before CAM appearance; then, MnSOD and Cu/ZnSOD activity increases after CAM occurs, induced

by salt [48,60]. CAT is not only responsible for the removal of $H_2O_2$, but presents diel fluctuations [58]. Non-enzymatic antioxidants constituted by low molecular metabolites include ascorbic acid (AsA), glutathione (GSH), carotenoids, γ-tocopherol, etc. [57]. In *M. crystallinum*, a transition from $C_3$ to CAM induced by $H_2O_2$ or salinity, α-tocopherol, polyamines and proline showed accumulation and performed a crucial role in preventing oxidative damage [31,61,62]. Likewise, during the $C_3$–CAM shift in *Guzmania monostachia* (a facultative CAM plant) induced by water stress under high light PFD, carotenoids were proven to play an important role in the ROS scavenging system [63].

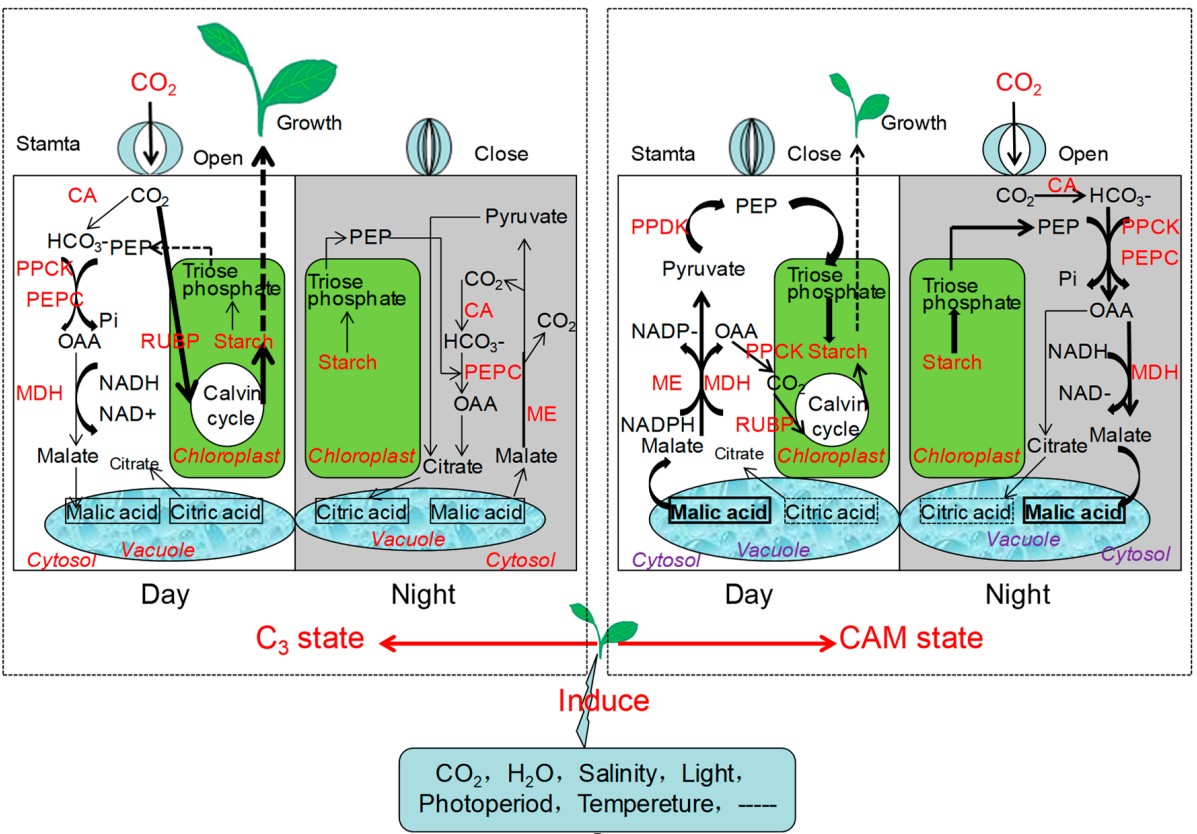

**Figure 1.** Physiological and Metabolic shift of the facultative CAM plants from $C_3$ to CAM photosynthesis. For the facultative CAM plants, $C_3$ state of photosynthesis is used to fix $CO_2$ under well conditions, when stomata open during the day and close at night. Once the facultative CAM plants stressed by abiotic stress (such as atmospheric $CO_2$ concentration, drought, salinity, photoperiod, light, etc.), the carbon assimilation pathway will be induced to the CAM mode, when $CO_2$ uptakes during the nighttime via open stomata and stores as organic acids (mainly malic acid). Then, organic acids are decarboxylated in the vacuoles during the daytime and $CO_2$ is refixed via Calvin cycle. CA, Carbonic anhydrase; PEP, Phosphoenol pyruvate; PEPC, Phosphoenolpyruvate carboxylase; PPCK, PEPC kinase; PPDK, Pyruvate phosphate kinase; ME, Malic enzyme; MDH, Malic dehydrogenase; OAA, Oxalacetic acid; RUBP, Ribulose-1,5-bisphosphate carboxylase; NAD, Nicotinamide adenine denucleotide; NADP, Nicotinamide adenine denucleotide phosphate.

## 5. Metabolic Mechanism during the $C_3$–CAM Shift in Facultative CAM Plants

CAM was controlled by the circadian clock and metabolites. Organic acids and carbohydrates are two kinds of major metabolites in facultative CAM plants [64]. Large diel changes in malic acid and transitory starch are important characteristics of CAM photosynthetic activity [65–67]. Plants, once stressed by adverse environmental factors, trigger their ABA- and $Ca^{2+}$-dependent signaling pathways, which leads to metabolite changes [28]. These diurnal changes in malic acid and/or citric acid regulate the expression of CAM

photosynthesis [68], but the acid accumulation requires metabolic reprogramming [69]. Metabolic fluxes modeling of a starch/sugar-malate cycle was established to test whether a $C_3$–CAM continuum exists in CAM evolution [70].

Carbohydrates, including starch and sucrose or hexose, are produced by the CAM pathway at a high energetic cost [65]. They are converted into PEP and provide substrates for the nocturnal $CO_2$ fixation [71]. In *M. crystallinum*, this transitory starch reserve is critical for CAM photosynthesis and the oscillation in starch levels between day and night (when the plant is in CAM mode) is 20% greater than in the $C_3$ mode [72,73]. Starch-deficient mutants of this plant are characterized by their inability to photosynthesize in the CAM mode, as these mutant plants are deficient in leaf starch. It was reported that there was a transitory starch breakdown by the hydrolytic pathway when plants were in the $C_3$ mode, but that a shift to the phosphorolytic pathway occurred during CAM photosynthesis [66]. In *M. crystallinum*, about half of the starch degraded during the night was used to supply PEP to ensure CAM photosynthesis, and the others were exported as soluble sugars used during plant respiration [74]. In another example, the Agave plant (*Yucca aloifolia*), a CAM plant, uses soluble sugars as a carbohydrates during CAM photosynthesis, while the $C_3$ plant *Y. filamentosa* is thought to rely on starch pools. However, the diploid hybrid species *Y. gloriosa*, a facultative CAM species, relies on starch for carbohydrates (like its $C_3$ parent *Y. filamentosa*), though many features are more similar to that of its CAM parent *Y. aloifolia* [75,76]. In facultative CAM plants, the proportion of $CO_2$ taken up between nighttime and daytime is influenced by developmental and environmental factors (such as drought, salinity and extreme temperature) [72,77]. Various studies have revealed that many species express a low degree of CAM photosynthesis and a peak in $C_3$-type isotope signatures. Generally, the $\delta^{13}C$ value has more $C_3$-type isotope signatures than CAM in facultative CAM plants. In *S. nuttallianum*, the $\delta^{13}C$ value was $-30.0‰$ but $\triangle H+$ was significant under controlled conditions [36]. *C. minor* can shift from $C_3$ to CAM photosynthesis easily and reversibly, and its $\delta^{13}C$ value was about $-21‰$ [78]. These studies showed that carbon was mostly fixed through $C_3$ photosynthesis in facultative CAM plants, and dark $CO_2$ fixation made up, at most, 30% [39].

## 6. Molecular Mechanisms during $C_3$–CAM Shift in Facultative CAM Plants

The photosynthetic shift from $C_3$–CAM is regulated by some enzymes and metabolite transporters, making this process a part of complex metabolic adaptations to environmental stress [22]. However, the relative expression of genes for enzymes and metabolites is strictly regulated by environmental factors. Genes involved in the shift of $C_3$–CAM may be controlled by a co-expressed circadian master regulator [79]. Therefore, describing the molecular mechanism behind the $C_3$–CAM shift is important to understand the evolution of CAM. Although every documented gene in the CAM also exists in the ancestral $C_3$ species, the timing and magnitude are different between $C_3$ and CAM species [80].

The PEPC gene is one of the most important genes in CAM and $C_4$ photosynthesis. In the past decades, the presence and function of this gene have been investigated in many CAM and facultative CAM plants. PEPC is a ubiquitous enzyme in higher plants and belongs to a small multigene family with several PEPC isozymes [52,81]. Plant PEPC activity is regulated by PEPC kinase (PPCK) [82]. Boxall et al. found that silencing PEPC (*Ppc*1) in the obligate CAM species *Kalanchoë laxiflora* can prevent $CO_2$ uptake and malate accumulation at night [83]. Here, the main molecular mechanisms that occur during the photosynthesis transition from $C_3$ to CAM in facultative CAM plants are as follows.

## 7. DNA Level Regulation

The photosynthesis shift from $C_3$ to CAM involved DNA hypermethylation [84]. For instance, variations in cytosine methylation were found in the *Ppc*1 promoter during the transition from $C_3$ to CAM in *M. crystallinum* [85].

## 8. Transcriptional Regulation

Between obligate CAM and $C_3$ photosynthesis plants, all of the CAM-related genes exist in ancestral $C_3$ species, but transcriptional regulation cascades are very important for the $C_3$–CAM transition, especially the expression of CAM-specific genes [86,87]. Some enzymes and genes that are involved in some facultative CAM plants during the photosynthesis from $C_3$ to CAM are shown in Table 1. The expression of CAM-specific PEPC, NAD-GAPDH and PPDK is important for the onset during the CAM induction; for example, PEPC mRNA accumulation occurs within 2–3 h, stressed by salinity in *M. crystallinum* or *K. blossfeldiana* [27,88,89]. Additionally, there are discrete changes observed in protein sequences [9,75,90]; the promoter regions of these CAM-specific genes contain GT motifs, which may function in light-responsive or ABA-mediated gene expression events [91,92]. The promoter sequences were different in the CAM-specific *PEPC* (*Ppcl*) and the $C_3$ "housekeeping" *PEPC*, that is the former containing TATA and CAAT box motifs, but there was absence in the latter [91]. For CAM-specific genes, there may be the same common cis-acting regulatory elements for regulating the stress-induced expression patterns in different facultative CAM plants. During CAM induction, the distal regions between $-977$ and $-721$ control the expression of *Ppcl* for salt-responsiveness, while the regions between $-735$ and $-675$ control the expression of NADGAPDH(*GapC*1) induced by salt [93]. Therefore, the *Ppcl* and *GapCl* promoters in the distal regions share multiple consensus binding sites of transcription factor which control the salt-inducible gene expression. In *Kalanchoë*, *PPC*1 is essential for the practice of CAM [83]. In facultative CAM plant *Talinum triangulare*, transcriptional regulation of the $C_3$–CAM transition revealed that the levels of the CAM-cycle enzyme transcripts are increased in response to drought stress [6]. In facultative CAM plants, during CAM induction induced by abiotic stress, a few transcription factors may control some transcriptional activation events to improve their tolerance [6,94].

Recently, several transcription factors (TF) families take part in regulating CAM induction by salinity or drought stress, such as AP2/ERF, MYB, WRKY, NAC, NF-Y, bZIP and McHB7 [93,95–98]. In *T. triangulare*, during CAM induction stressed by ABA, transcription factors such as HSFA2, NF-YA9 and JMJ27 were identified as regulators for the CAM induction [21]. Cushman and Bohnert [99] demonstrated that one factor (designated *PCAT*-1) binds in the *Ppcl* promoter, and the *PCAT*-1 expressed is abundant and may play an important role in the assembly of active transcription complexes during the photosynthesis shift from $C_3$ to CAM.

**Table 1.** Some enzymes and genes involved in some facultative CAM plants during the photosynthesis shift from $C_3$ to CAM.

| Enzyme | Gene | Source/Species | Inducer | References |
|---|---|---|---|---|
| Phosphenolpyruvate carboxylase | *Ppc*1 | *M. crystallinum* | salt, ABA, drought, cytokinin | [25,100] |
| | *Kb*-1, *Kb*-2 | *K. blossfeldiana* | short-day, drought | [101] |
| | *Ppc* 3 | *T. triangulare* | ABA | [21] |
| | $C_3$-type PEPCs | *C. minor* | drought | [102] |
| Alpha Carbonic Anhydrase 1 | *ACA1* | *T. triangulare* | ABA | [21] |
| Beta Carbonic Anhydrase 5 | *BCA5* | *T. triangulare* | ABA | [21] |
| Malic Enzymes | *MEs* | *T. triangulare* | ABA | [21] |
| PEPC Kinase | *PPCK1* | *M. crystallinum* T. triangulare | Salt ABA | [21,103,104] |
| Pyruvate orthophosphate dikinase(PPDK) | *Ppdk*1 | *M. crystallinum* T. triangulare | salt, ABA | [21,105] |
| Enolase | *Pgh*1 | *M. crystallinum* | salt, drought, cold, hypoxia, ABA, 6-BA | [106] |

**Table 1.** *Cont.*

| Enzyme | Gene | Source/Species | Inducer | References |
|---|---|---|---|---|
| phosphoglyceromutase (PGM) | *Pgm*1 | *M. crystallinum* | salt, drought, ABA, 6-BA | [107] |
| GAD-Glyceraldehyde 3-phosphate dehydrogenase (GAPDH) | *Gap*C1 | *M. crystallinum* | salt | [93,108] |
| NADP-Malic enzyme | *Mod*1 | *M. crystallinum* T. triangulare | salt ABA | [21,109] |
| | *Mod*4 | *T. triangulare* | ABA | [21] |
| NADP-Malate dehydrogenase | *MDH*1 | *M. crystallinum* | salt | [80] |
| NAD-Malate dehydrogenase | *MDH*2 | *M. crystallinum* | salt | [110] |
| $H^+$-ATPase. c subunit | *Atpvc* | *M. crystallinum, K. daigremontiana* | salt, ABA, light | [111–113] |
| $H^+$-ATPase, E subunit | *AtpvE* | *M. crystallinum* | salt | [114] |
| SNF1 kinase | *MK*9 | *M. crystallinum* | salt | [115] |
| RNA-binding protein | *Rbp*1 | *M. crystallinum* | salt | [116] |
| Ribosome inactivating proteins | *Rip*1 | *M. crystallinum* | salt | [117] |

## 9. Post-Transcriptional Regulation

Post-transcriptional regulations also take part in regulating CAM expression, such as that *Ppc*1 may facilitate long-term CAM build up by increasing mRNA stability in *M. crystallinum* during salt stress [118,119]. In *M. crystallinum,* many cDNA libraries were constructed with different tissues and stress treatments [120]. Large-scale steady-state mRNA abundance was found to change significantly during CAM induction in plants subjected to salinity-related stress [121]. There was at least one CAM-specific PEPC isoform (*Ppc*1) responsible for $CO_2$ fixation during the night, which is more abundant than during the day and is responsible for CAM expression [75,122]. During the photosynthesis shifts from $C_3$ to CAM in *T. triangulare*, PEPC isoform 1(*Ppc*1) and isoform 3 (*Ppc*3) transcript abundance increased and PPDK transcripts started to accumulate after 80 min, but the Alpha Carbonic Anhydrase 1 (*ACA*1) transcript decreased [21]. When *M. crystallinum* was treated for seven days with salt, *Ppc*1 and *PPCK*1 were up-regulated in guard cells [104].

Omics approaches were used by many scientists to explore the underlying molecular mechanism during the transition from $C_3$ to CAM [6,123]. Among monocot species, the partial transcriptomes or genomes in the genus *Phalaenopsis* have been characterized [124]. The rapid reversible $C_3$–CAM shift in the genus *Clusia* is based on the $C_3$ isoform of *PEPC* (a housekeeping gene) [102]. The post-transcriptional regulation of photosynthetic genes is a key driver of $C_4$ leaf ontogeny, determined by using exon–intron split analysis [125]. In order to explore the complex regulatory mechanisms, Heyduk et al. elaborated on a comparative analysis with closely related $C_3$ and CAM species [86]. Aside from PEPC, other enzymes are also important for CAM photosynthesis, which are the products of isogenes. Other studies showed that following CAM induction by salinity-related stress, several CAM-related glycolysis/gluconeogenesis genes showed increasing transcript abundance [106–108]. In order to identify the genes involved during the transition from $C_3$ to CAM in *M. crystallinum*, genomics and transcriptomics analyses were combined; twenty genes encoding six main enzymes were identified and one of four MDH genes presented a specific function in CAM photosynthesis [126]. Many CAM-related starch synthesis/degradation genes have been identified in *M. crystallinum*. During the photosynthesis shift from $C_3$ to CAM, ADP glucose pyrophosphorylase small (*Agp*1 and

*Agp*2) and large subunit (*Agp*3) catalyzed the starch biosynthesis and showed an increase in mRNA expression; additionally, there were three genes including *AmyA*1 (α-amylase isogene), *AmyB*1 and *AmyB*2 which exhibited increased remarkable mRNA abundance during nocturnal starch degradation [121].

Many microRNAs (miRNAs) also play a regulatory role in CAM photosynthesis in the leaves of obligate CAM species, such as *Ananas comosus* [127]. MiRNAs in *M. crystallinum* seedlings under salinity-related stress were analyzed by RNA sequencing and were found to be involved in the post-transcriptional regulation of salt tolerance [128]. Hu et al. identified some miRNAs that were involved in the regulation of CAM in *Kalanchoë* and found that the miR530-TZPs module regulates CAM-related gene expression [129].

## 10. Protein Level Regulation

Regulatory proteins exist in $C_3$, CAM and $C_4$ species and are essential to the $C_3$–CAM photosynthesis shift in facultative CAM plants. In *M. crystallinum,* salt-stressed, heat shock proteins and early light-inducible proteins were found increased in the cDNA libraries, and a few proteins were found increased in guard cells but decreased in mesophyll cells [130,131]. There were seven proteins with increased expression and four proteins with decreased expression in *M. crystallinum* induced by salt [98]. Additionally, new major phosphorylation events during the transition from $C_3$ to CAM stressed by salt were identified and characterized using proteomics and phosphoproteomics [132].

## 11. Implications in Horticultural Crops

CAM photosynthesis enables plants to assimilate carbon under environmental stress conditions. In contrast to $C_3$ and $C_4$ plants, obligate CAM plants have a higher transpiration efficiency and a lower photosynthetic rate; hence, obligate CAM plants often grow more slowly than their $C_3$ and $C_4$ counterparts [133]. Thus, CAM is not the best choice for highly productive plants [14]. However, facultative CAM plants can function in a $C_3$ mode to increase photosynthetic rates and growth when there are no physical limitations present, and shift to the CAM mode to decrease water loss and overcome environmental stressors. For facultative CAM plants, the photosynthesis switching between $C_3$ and CAM has important ecological implications. Therefore, engineering CAM into $C_3$ crops can improve their WUE and sustain crop productivity in hot and dry climates [15,16,134].

CAM plants are widely distributed within the plant kingdom, i.e., 343 genera in 34 families, approximately 6.5% of flowering plant species [135]. In fact, most crops practice either $C_3$ or $C_4$ photosynthesis, but not CAM photosynthesis, such as wheat, rice and maize, which have higher production. However, many horticultural plants belong to obligate CAM plants or facultative CAM plants; some of them (such as pineapple, an obligate CAM plant, practice a facultative $C_3$/CAM metabolism in the first 2 months of growth) can also be very productive when cultivated under well-conditions [55]. However, there was no exact number in facultative CAM plants. Winter (2019) reported that facultative CAM plants exist in at least 15 families, and he thought there may be over 1000 facultative CAM species in Aizoaceae alone [9]. Many orchids (*Dendrobium* spp., *Oncidium* spp. and *Phalaenopsis* spp.) [52,124,136,137], with higher ornamental values or medicinal values, were identified as facultative CAM plants since they could switch the pathway between $C_3$ and CAM according to the environmental condition. For example, *D. officinale,* an important traditional herb with higher commercial value in China, uses the facultative CAM pathway to increase its drought tolerance [138]. Many species of Portulacaceae belong to facultative CAM plants [9]. *Jatropha curcas*, an oil crop, could also practice CAM photosynthesis for survival in response to environmental stress [139]. Thus, this plastic photosynthetic adaptation results in important implications for many horticultural crops.

## 12. Future Perspectives

For the facultative CAM plants, the photosynthesis switches from $C_3$ to CAM have important ecological implications. Facultative CAM can prevent $CO_2$ loss and favors plant

growth and reproduction in responses to environmental stress. During the C$_3$–CAM shift in facultative CAM plants, although there are some progresses in anatomy, physiological, metabolic and molecular properties of facultative CAM plants, in the near future, there are still some works that should be carried out. First, the argument on anatomical variations should be comprehensive and ongoing, and studied with more genera that include obligate C$_3$, facultative CAM and obligate CAM plants. For example, many characters and mechanisms can be explored in the genera *Clusia* (Clusiaceae), *Dendrobium* and *Oncidium* (Orchidaceae) and *Yucca* (Asparagaceae), which are well known for containing the obligate C$_3$, facultative CAM and obligate CAM plants in a single genus [19,76,102,136]. Second, molecular mechanisms underlying the transition from C$_3$ to CAM in plants is still limited. Recently, omics approaches including transcriptomic, genomic, proteomic, metabolomics and ionomics were used frequently to reveal the molecular changes during the C$_3$–CAM transition (See Table 2). For example, altered gene regulatory networks and expression profiles were found in the transition from C$_3$ to CAM in *Erycina* (Orchidaceae) and *Yucca* (Asparagaceae) [76,86], which will benefit the clarification of the key molecular switches underlying this transition of C$_3$ to CAM in facultative CAM plants. Third, understanding the functional genomics of CAM plants is important to elucidate the relationship between genotype and phenotype [9]. Synthetic biology toolboxes, such as the CRISPR/Cas 9 system which was confirmed to be effective for genome editing in *K. fedtschenkoi*, will accelerate the ongoing research about CAM and the C$_3$ to CAM transition mechanisms [17,140]. Assisted with all this prior knowledge, we should obtain more genetic transformation information on facultative CAM plants and transfer more CAM-related genes into C$_3$ types of horticultural crops in future.

**Table 2.** Omics approaches involved to reveal the molecular changes during the C$_3$–CAM transition in facultative CAM plants.

| Omics Approaches | Source/Species | Photosynthesis Type | Year | Reference |
|---|---|---|---|---|
| Proteomics, Metabolomics | *M. crystallinum* | facultative CAM plants | 2013 | [141] |
| Transcriptomics | *M. crystallinum* | facultative CAM plants | 2015 | [142] |
| Metabolomics | *M. crystallinum* | facultative CAM plants | 2015 | [143] |
| Transcriptomics | *M. crystallinum* | facultative CAM plants | 2015 | [144] |
| Proteomics, Ionomics | *M. crystallinum* | facultative CAM plants | 2016 | [145] |
| Transcriptomics, Metabolomics | *T. triangulare* | facultative CAM plants | 2016 | [6] |
| Transcriptomics | *D. catenatum* | facultative CAM plants | 2016 | [124] |
| Transcriptomics | *Agave* (CAM), *Polianthes* (weak CAM), *Manfreda* (CAM), *Beschorneria* (weak CAM) | CAM plants | 2018 | [95] |
| Transcriptomics | *D. catenatum* | facultative CAM plants | 2018 | [137] |
| Transcriptomics | *Erycina pusilla* (CAM), *Erycina crista-galli* (C3), | CAM plants, C$_3$ plants | 2019 | [86] |
| Transcriptomics, Metabolomics | *T. triangulare* | facultative CAM plants | 2019 | [21] |

**Table 2.** *Cont.*

| Omics Approaches | Source/Species | Photosynthesis Type | Year | Reference |
|---|---|---|---|---|
| Metabolomics Transcriptomics | *Y. gloriosa* (C$_3$+ CAM), *Y. filamentosa* (C3), *Y. aloifolia* (CAM) | facultative CAM plants, C$_3$ plants, obligate CAM plants | 2019 | [76] |
| Genomics | *Sedum album* | facultative CAM plants | 2019 | [123] |
| Transcriptomics | *M. crystallinum* | facultative CAM plants | 2020 | [104] |
| Proteomics, Metabolomics | *M. crystallinum* | facultative CAM plants | 2021 | [131] |
| Proteomics | *M. crystallinum* | facultative CAM plants | 2021 | [98] |
| Proteomics, Phosphoproteomics | *M. crystallinum* | facultative CAM plants | 2022 | [132] |
| Transcriptomics | *Tamarix ramosissima* | facultative CAM plants | 2022 | [146] |
| Transcriptomics Genomics | *M. crystallinum* | facultative CAM plants | 2022 | [126] |
| Transcriptomics | 11 species of Agavoideae | facultative CAM plants, C$_3$ plants, obligate CAM plants | 2022 | [147] |

**Author Contributions:** Conceptualization, Z.Z. and S.Q.; writing and revision of the manuscript, S.Q.; revision and editing of the manuscript, K.X.; Y.Y. and Q.W. All authors have read and agreed to the published version of the manuscript.

**Funding:** This work was funded by the National Natural Science Foundation of China (31560567), National Natural Science Foundation of Guangxi (2020GXNSFAA297260), Start-up Fund of Innovation Team of Guangxi Academy of Sciences for Innovation and Utilization of Germplasm in Horticultural Crops (CQZ-E-1919), Fundamental Research Fund of Guangxi Institute of Botany (23011) and the fund of Guangxi Key Laboratory of Plant Functional Phytochemicals and Sustainable Utilization (ZRJJ2022-5 and ZRJJ2023-1).

**Conflicts of Interest:** The authors declare no conflict of interest.

**Abbreviations**

| | |
|---|---|
| ABA | Abscisic acid |
| CAM | Crassulacean acid metabolism |
| APX | Ascorbate peroxidase |
| CAT | Catalase |
| H$_2$O$_2$ | Hydrogen peroxide |
| MDH | Malate dehydrogenase |
| ME | Malic enzyme; |
| NO | Nitric oxide |
| PEPC | Phosphoenolpyruvate carboxylase |
| PEPCK | PEP carboxykinase |
| POD | Peroxidase |
| PFD | Photon flux density |
| Rubisco | Ribulosebisphosphate carboxylase/oxygenase |
| SOD | superoxide dismutase (SOD) |
| TF | Transcription factors |
| TZPs | Tandem zinc knuckle/PLU3 domain encoding genes |
| WUE | Water-use efficiency |

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
