# Peer review of "Mechanisms Underlying the C3–CAM Photosynthetic Shift in Facultative CAM Plants"

_horticulturae, doi:10.3390/horticulturae9030398_

Round 1

Reviewer 1 Report

Regarding this manuscript (´Mechanisms and ecological implications underlying the C3-CAM photosynthetic shift in facultative crassulacean acid metabolism plants`), I suggest that the authors consider the following concerns:

1) I suggest that the authors increase the abstract length by adding more information about the mentioned progress in research involving anatomical, physiological, metabolic, and molecular properties. I believe at least 200 words would be interesting

2) After reading this manuscript, I noticed that some recent articles on this topic were not cited. Please see the references (as examples):

´Metabolic Modeling of the C3-CAM Continuum Revealed the Establishment of a Starch/Sugar-Malate Cycle in CAM Evolution` ( https://doi.org/10.3389/fpls.2020.573197 )

´CAM photosynthesis: the acid test` ( https://doi.org/10.1111/nph.17790 )

3) In the introduction, I recommend authors provide more recent references for some statements and sentences. For example, in the sentence ´However, the complex shift mechanism between C3 photosynthesis and CAM in facultative CAM plants in response to environmental stress remains poorly understood, particularly concerning the genetic changes required for the progression and reversion of this shift`, a reference from 2010 (instead of a recent one) was cited.

4) I did not understand why the following paragraph comes after the paragraph where the authors mention the objective of the article: ´CAM photosynthesis enables plants to assimilate carbon under environmental stress 50 conditions.... Facultative CAM plants can struggle with variable environments through the facultative CAM mode[17]` (lines 50 to 56)`

5) I think the introduction is too short. To me, that sounds almost like an abstract in length

6) In Table 1, all plant species names should be italicized

7) I recommend rewriting the following sentence (since it is identical to the sentence in the original article):

´ were up-regulated in guard cells after seven days of salt treatment, indicating that guard cells themselves can shift from C3 to CAM`

8) Both the title of the article and the abstract increases the reader expectation towards ecological implications. However, the topic ´ Ecological implications` (lines 298 to 312) is too short.

9) In the conclusion, the word 'Further more` should be revised (line 329)

10) The authors may explain what criteria they used to write the section 'Protein level regulation` (lines 286 to 297), considering that several proteomic and interesting studies on this topic were not cited.

Some of examples of interesting proteomic studies on this review topic that were not cited:

´Proteomics and phosphoproteomics of C3 to CAM transition in the common ice plant` (https://doi.org/10.1016/bs.mie.2022.06.004)

´ Comparative proteomics of Mesembryanthemum crystallinum guard cells and mesophyll cells in transition from C3 to CAM ` ( https://doi.org/10.1016/j.jprot.2020.104019 )

11) In the topic ´ Future perspectives`, the authors cited some omics approaches. But they could also consider other omics approaches (e.g., Metabolomics and Ionomics) and also provide a table on studies involving such omics studies to cover as many articles as possible. I cannot see how to summarizee the progress in molecular or biochemical properties of facultative CAM plants between C3 and CAM photosynthesis without considering as many papers as possible involving omics approaches (proteomics, transcriptomics, etc).

12) The authors mentioned that ´During the C3-CAM shift in facultative CAM plants induced by environmental stress, the antioxidative stress response increases[46]` (lines 144 and 145). Given the importance of antioxidants in multiple physiological and biological processes in plants, I believe the authors could discuss this topic better by considering both enzymatic antioxidants and non-enzymatic antioxidants.

Author Response

Thank you for your advice. Please see the reply in appendix.

Reviewer 2 Report

This manuscript is a review of the current state of research on “Mechanisms and ecological implications underlying the C3-CAM photosynthetic shift in facultative crassulacean acid metabolism plants.

There have been 10 reviews on the subject in past 2 years. 5 in 2022 alone. The review does not provide any new information and has just repurposed language. I think this manuscript could benefit from a better framing and synthesis of current studies into a meta-analysis.

Line 276 is post transcriptional regulation – check for the accuracy. “While switching from C3 to CAM, 2 cDNA from starch synthesis/degradation genes were upregulated. When in CAM mode starch is being degraded not synthesized.”

"Protein level regulation" line 286 - cDNA upregulation is not protein level regulation.

Line 290 has no purpose.

Figure 1 needs clarification. 

The whole paper seems to be reworded statements from previous articles joined together with no clear message other than in a "it has been shown" vernacular sometimes with incorrect conclusions.

Manuscript needs to be reviewed by a native English speaker again as there are spelling and grammatical errors.

Author Response

Thank you for your advice. Please the response in appendix.

Reviewer 3 Report

The article dealts with fundamental problem in plant physiology  -a shifts between  main C3 - and SAM-metabolism and may be published in the journal. Although principial news are not  in the text, nevertheless authors concentrate the science audience as a whole on the problem and remind about the search in the direction. Conservative academic style is present here as in old journals of MDPI.

Author Response

Thank you for your advice. Please see the response in appendix.

Reviewer 4 Report

The paper is very interesting and I hope it will be very good in the field of research.

 I am agree with this paper to be published in this form.

Author Response

Thank you for your advice. According comments of other reviewers, we did some revision. Please see the revised version and response in appendix.

Round 2

Reviewer 1 Report

I recommend that the authors improve the topic on reactive oxygen species and antioxidants:

The sentence ´are always formedand enzymatic antioxidants and non-enzymatic antioxidants are involved to protect the plants in response to environmental stress` is grammatically incorrect.

The authors should provide a reference to the phrase: 'Non-enzymatic antioxidants, including ascorbic acid (AsA) and reduced glutathione (GSH) are also crucial for H2O2 removal in the ascorbate–glutathione cycle (AGC)`.

Furthermore, the authors could better explain the function of non-enzymatic antioxidants in general by citing interesting reviews.

Author Response

Thank you very much for you advice. We have made some revise, please see the appendix.

Reviewer 2 Report

na

Author Response

Thank you many. We checked the manuscript and do some revision about the language. Please see the revised version.